# Exploring issues surrounding mental health and wellbeing across two continents: A preliminary cross-sectional collaborative study between the University of California, Davis, and University of Pretoria

**Munashe Chigerwe**[1]*, **Dietmar E. Holm**[2], **El-Marie Mostert**[3], **Kate May**[2], **Karen A. Boudreaux**[4]

1 Department of Veterinary Medicine and Epidemiology, School of Veterinary Medicine, University of California Davis, Davis, California, United States of America, 2 Department of Production Animal Studies, Faculty of Veterinary Science, University of Pretoria, Onderstepoort, Pretoria, South Africa, 3 Department for Education Innovation, Faculty of Veterinary Science, University of Pretoria, Onderstepoort, Pretoria, South Africa, 4 Dean's Office, School of Veterinary Medicine, University of California Davis, Davis, California, United States of America

* mchigerwe@ucdavis.edu

**Data Availability Statement:** The data presented n our study potentially contain sensitive identifying

## Abstract

Mental health and wellness research continue to be a topic of importance among veterinary students in the United States of America (US). Limited peer reviewed literature focusing on South African veterinary students is available. South African veterinary medical students might benefit from approaches to improve mental health and wellness similar to those recommended in the US. However, these recommendations may not address the underlying risk factors for mental health and wellness concerns or mismatch resources available to South African veterinary medical students. The purpose of this collaborative study was to compare the mental health and wellness among veterinary students enrolled at the University of California, Davis (UCD), and the University of Pretoria (UP), the only veterinary school in South Africa. Our primary research question was; Are the measures of mental health and wellness for students at similar stages in the veterinary curriculum different between the two schools? We hypothesized that mental health and wellness as determined by assessment of anxiety, burnout, depression, and quality of life between the two schools is different. A cross-sectional study of 102 students from UCD and 74 students from UP, at similar preclinical stages (Year 2 for UCD and Year 4 for UP) of the veterinary curriculum was performed. Anxiety, burnout, depression, and quality of life were assessed using the Generalized Anxiety Disorder (GAD-7), Maslach Burnout Inventory (MBI), Patient Health Questionnaire (PHQ-9), and Short Form-8 (SF-8), respectively. Students from both schools had moderate levels of anxiety, high levels of burnout, mild to moderate levels of depression, poor mental health, and good physical health. Our results suggest that similar mental health and wellness concerns in South African veterinary students is comparable with concerns in veterinary medical students in the US. Recommendations and resources to improve

personal information. We have provided contacts to which data requests may be sent. The contacts are the respective institutional representatives for the ethics committees: University of California Davis: Office of Research IRB Administration 1850 Research Park Drive Davis, CA 95618-6153 Phone: (530) 754-7679 Fax: (530) 754-7894 Email: ORExecutiveMgtAsst@ad3.ucdavis.edu University of Pretoria: Deputy Dean, Postgraduate and Research Ethics Faculty of Humanities Email: PGHumanities@up.ac.za.

**Funding:** The study was funded by the University Capacity Development Grant of the Department of Higher Education and Training (DHET) of South Africa, and the UC Davis Faculty Discretionary Support funds. There was no additional external funding received for this study.

**Competing interests:** The authors have declared that no competing interests exist.

mental health and wellness in US veterinary medical students might be applicable to South African veterinary medical students.

## Introduction

Several universities educating veterinarians in the United States have recently undergone curricular revision. The University of California, Davis, School of Veterinary Medicine (UCD) completed its curricular revision for its Doctor of Veterinary Medicine (DVM) in 2011, graduating its first class from the revised curriculum in 2015 [1]. The UCD's revised integrated block design curriculum was anticipated to reduce student and faculty stress and burnout by focusing learning material to fundamental "Day One Skills" necessary for graduates [1]. The revised curriculum was student-centered, inquiry-based, and included teaching and assessing methods that promoted critical thinking, such as case analysis and veterinary practice applications [1]. The University of Pretoria's Faculty of Veterinary Science (UP) completed a major curricular revision in 2016 [2]. While the UP's driving forces for change were programmatic and educational, the UCD identified student and faculty stress and burnout as an additional impetus for curricular change. While the UP did not specifically identify mental health and wellness as a factor for change, the concerns for student mental health and wellness recently became prominent in the South African higher education sector [2]. The concerns for mental health and wellness were compounded by student protests prompted by persistence of powerful symbols of colonial history of tertiary institutions, lack of gender and racial diversity, and financial pressures on students during the curricular revision process [2]. Therefore, student mental health and wellness among veterinary students is associated with design, modification or change of curriculum. The similarities between drivers for curricular changes between UCD and UP programs included emphasis on core learning material for entry level veterinarians (Day One Skills), and recognition of the influences of external forces such as licensing accrediting bodies, and other stakeholders [1, 2].

There is a significant body of literature on mental health and wellness available for students in US veterinary schools [3–5]. In one study across three consecutive semesters at a US school, proportions as high as 49%, 65%, and 69%, respectively, of veterinary students reported depression levels above the clinical cut-off [4]. Furthermore, a cross-sectional study of 1,245 students from 33 US veterinary colleges reported that 66% of students had symptoms of mild to moderate depression, and these levels were higher compared to college students in other programs (41%) or medical students (23%) [5]. Factors associated with increased scores of anxiety and depression among US veterinary medical students include transitional stress (difficulty fitting in), academic stress (heavy workload), and homesickness [3, 4]. Female veterinary students have been reported to have higher scores for depression compared to males [4, 5].

In contrast, little peer reviewed information is available describing factors associated with mental health and wellness among South African veterinary medical students. However, factors associated with anxiety and depression in non-veterinary students and the general South African adult population include cultural differences [6, 7] and race [6, 8, 9]. Furthermore, social, and economic factors such as higher number of household members, lower education attainment, gender, lower social status, multiracial race, and less income stability also have been associated with depression and anxiety in South African adults [10]. Thus, some factors that are negatively associated with optimal mental well-being among veterinary students on the two continents might be different. For instance, anecdotal perceived important stressors negatively associated with optimal mental well-being in South African students include

loneliness due to the geographical isolation of the veterinary school, and electrical power failures (thereby reducing study time). In contrast, perceived poor health and unclear curriculum expectations are major stressors negatively associated with optimal mental well-being in US veterinary students [3–5]. Furthermore, the resources available to support veterinary students' mental health and wellness are different between US and South African schools of veterinary medicine. Despite these potential differences in resource availability, South African veterinary students may adopt approaches to improve mental health and wellness similar to those recommended in the US. This is because some of the perceived stressors associated with poor mental health in veterinary students such as increased academic workload, and lack of racial diversity in class composition are common to South African and US veterinary schools. However, these recommendations may not address the underlying risk factors for mental health and wellness concerns or mismatch available resources in South Africa. Consequently, collaboration between schools of veterinary medicine on two different continents will not only promote collaboration amongst educational researchers in veterinary medicine but enhance communication and provide research data that advances the knowledge of mental health and wellness issues in veterinary students.

## Current mental health and wellness support for veterinary students

The University of California Davis and UP provided the following onsite mental health and wellness programs (excluding campus-wide, state, or national resources) at the time of study:

### University of California Davis

The UCD wellness program had one full time and one part-time professional counselors on staff to provide psychotherapy. The Wellness Center was equipped with individual quiet rooms for sleeping, resting, and relaxation massage chairs. The Wellness Center was located within a walking distance from lecture rooms and the teaching hospital. A wellness initiative referred to as 'Wake up for Wellness' was held several times a semester to promote self-care and wellness, boost morale, and built a sense of community. A veterinary student-run Health and Wellness Club provided opportunities for self-care and promoted mental, physical and emotional health, and wellbeing through activities such as organizing presentations and webinars related to mental health and wellness, and physical activities such as yoga (along with faculty), meditation, wine-tasting, cooking, and hiking.

### University of Pretoria

The UP had a one full-time faculty advisor providing student support on issues such as life-balancing, time management, motivation, and stress management. A part-time professional counselor was available to provide psychotherapy to veterinary students. A part-time physician (general medicine) and a part-time nurse were also available through the Faculty Medical Services. All professionals providing mental health and wellness support to the students could refer cases to a comprehensive campus service (University of Pretoria Counselling Services). The Onderstepoort Paraveterinary and Veterinary Student Committee, which was a professionally trained peer group provided opportunities and support for students through workshops and presentations on approaches to manage stress, anxiety, and depression.

### Current study

This study provides an opportunity to examine mental health and wellness issues in programs of veterinary medicine in two countries potentially providing more information about the

drivers and risk factors of mental health and wellness in schools of veterinary medicine. Specifically, this study focuses on the mental health and wellness aspects of students in terms of anxiety, burnout, depression, and quality of life. This preliminary, collaborative, cross-sectional study compared the mental health and wellness among students enrolled at UCD, United States of America, and UP, South Africa. Our primary research question was; Are the measures of mental health and wellness for students at similar stages in the veterinary curriculum different between the two schools? We hypothesized that mental health and wellness as determined by assessments of anxiety, burnout, depression, and quality of life would be different between the two schools.

## Study design methods

### Sampling and data collection

Sampling consisted of a non-probability sampling method via the use of a voluntary and convenient sample of students enrolled at the end of their second and fourth year of the DVM programs at UCD, and UP, respectively. While the DVM program is a 4-year curriculum at UCD requiring a bachelor's degree for a student to be eligible, the Bachelor of Veterinary Science (BVSc) program is a 6-year curriculum at UP where students complete their first year of basic natural sciences education at the Hatfield campus in the Faculty of Natural and Agricultural Sciences, then transfer into the veterinary portion of the curriculum to complete the second through sixth years of education at the Onderstepoort campus in the Faculty of Veterinary Science. The UCD begins its academic year in August, while UP commences in February. Data were collected in May 2018 for UCD and in November 2018 for UP, prior to the onset of the end of year examinations. Both data collection periods utilized the Qualtrics online survey tool where students were informed that the data collected were confidential but not anonymous as responses were tracked via individual emails to ensure each student responded only once to the survey. This data collection tool also allowed for follow-up reminders. Consent by students to participate in the study was requested via email. Consent by students was electronically written. The study was approved by the UCD Institutional Review Board (Decision HRP 503) and the UP-Research Ethics Committee (#GW20181001).

### Instrumentation

The following instruments were used to assess anxiety, burnout, depression, and quality of life of students at both schools.

### Anxiety

The Generalized Anxiety Disorder (GAD-7) [11] was a self-administered patient questionnaire used as a screening tool and severity measure for generalized anxiety disorder. The GAD-7 consisted of seven questions where students indicated the frequency they experienced designated problems. Scores were calculated by assigning a value of 0, 1, 2, or 3, to the response categories of 'not at all', 'several days', 'more than half the days', and 'nearly every day', respectively, and the sum was calculated for the seven questions for a total score ranging from 0 to 21. Scores of 5, 10, and 15 were considered the cut-off points for mild, moderate, and severe anxiety, respectively. The GAD-7 had established internal and test-retest reliability and criterion, construct, factorial and procedural validity with sensitivity and specificity scores of 89% and 82%, respectively [11]. The GAD-7 was selected to assess anxiety over other instruments, including the Mental Health Inventory [12], the Zung Self-Rating Anxiety Scale [13], the State-Trait Anxiety Inventory [14], and the Trimodal Anxiety Questionnaire [15] because of its efficiency (short length of 7 items), good reliability, criterion construct, factorial and procedural validity [11].

## Burnout

The Maslach Burnout Inventory (MBI) [16] was an instrument that measures individual burnout levels through the subscales of emotional exhaustion, depersonalization, and personal accomplishment which are identified as the key aspects to burnout, thereby providing a context for why burnout has possibly occurred [16]. The emotional exhaustion subscale assessed feelings of overextension and exhaustion from schoolwork with total scores of ≤ 18, 19–26 and ≥27 indicating low, average, and high levels of burnout, respectively. The depersonalization subscale measured an impersonal response to other students with total scores of ≤ 5, 6–9 and ≥10 indicating low, average, and high levels of burnout, respectively. Total personal accomplishment subscale measured students' feelings of competence and success with scores of ≥40, 39–34 and ≤33 indicating low, average, and high levels of burnout, respectively.

## Depression

The Patient Health Questionnaire (PHQ-9) [17] included a depression module that is part of the self-administered version of the PRIME-MD (Primary Care Evaluation of Mental Disorders) diagnostic instrument for common mental disorders. The PHQ-9 scored nine depressive symptom criteria present during the previous 2 weeks as 0 (not at all) to 3 (nearly every day) resulting in a severity measure ranging from 0 to 27. Total scores of 0–4, 5–9, 10–14, 15–19, and 20–27, indicated none, mild, moderate, moderately severe, and severe levels of depression, respectively. The PHQ-9 had demonstrated construct validity for use in primary care yielding a test-retest reliability of 0.84, sensitivity of 88% and specificity of 88% for major depression. The PHQ-9 was used to monitor the severity of depression and response to treatment as it is not a screening tool for depression. While several instruments are available to assess depression, including the Zung Depression Rating Scale [18], various versions of the Beck Depression Inventory [19, 20], and the Center for Epidemiological Studies Depression Scale [21], the PHQ-9 was selected for our study because of its ability to assess severity of depression, along with its acceptable reliability and validity coefficients [17].

## Quality of life

The Optum Short Form-8 Health Survey (SF-8) was an 8-item self-reported Likert scale instrument that assessed health related quality of life with established reliability and validity for the health domain scales of general health perception (GH), physical functioning (PF), role limitations due to physical health problems (role physical, RP), bodily pain (BP), energy/fatigue (vitality, VT), social functioning (SF), role limitation due to emotional problems (role emotional, RE), and psychological distress and well-being (mental health, MH) [22]. Two summary measures were determined, namely the physical component summary (PCS) and the mental component summary (MCS). Alternate-form reliability for a 4-week recall for the subscales ranged from 0.70 to 0.88. Responses were standardized into scores for the eight dimensions using Optum Pro CoRE software (Optum Pro CoRE, Eden Prairie, MN). Scores below 50 points (mean score = 50, standard deviation = 10) correspond to deviations from normality and indicated a poorer quality of life, whereas scores above 50 points represented a better quality of life than that of the average adult American population [22].

## Data analysis

Descriptive statistics were reported for response rate, gender, marital status, and age. Normality check of the data was performed by the Shapiro-Wilk test. When data were normally distributed, mean ± standard deviation (SD) was reported, whereas median and 95% confidence

interval (95% CI) was reported for not normally distributed data. Overall reliability (internal consistency) of each instrument was performed by calculation of Cronbach's alpha coefficient. Cronbach's alpha values of $\geq$ 0.7 were considered to indicate acceptable reliability [23]. A one-way between-groups multivariate ANOVA (MANOVA) was performed to determine significant differences between the scores for the survey instruments between UCD and UP students. Independent variables considered were group (UCD or UP), gender, marital status, and age. Correlations between the subscales for the survey instruments were also calculated. Commercial software was used for analyses (GraphPad Prism Version 8.1.2, GraphPad Software, San Diego, CA; JMP Pro Version 14, SAS Institute Inc., Cary, NC). $P <0.05$ was considered significant.

## Results

Response rates were 68% and 51% for UCD and UP, respectively. A total of 102 students (101 females and 1 male) from a class of 150 at UCD completed all instruments. Results of the 1 male student enrolled from the UCD cohort was excluded from further analysis of comparisons between the 2 schools. Seventy-four (52 females, 21 males, 1 non-binary) students from a class of 145 from UP, completed all instruments. Sixty-one (82%) and 13 (18%) students were aged 20–25, and 26–30 years respectively, at UP. At UCD, 101 (99%) and 1 (1%) students were aged 20–25, and 26–30 years, respectively. Further analysis of age comparisons between the schools excluded the 1 student from UCD aged 26–30. Three students (4%) at UP indicated they had a spouse/partner, whereas 70 (96%) were single. Nineteen (19%) students at UCD indicated they had a spouse/partner whereas 83 (81%) were single. The ethnic composition for UCD students was 82 White/Caucasian, 30 Asian, 24 Multi-ethnic, 7 Hispanic, Latino or Spanish origin, and 1 African American/Black. The ethnic composition of UP students was 104 White/Caucasian, 17 Black African, 17 Indian, and 7 colored (in the South African population the colored ethnicity is synonymous to the mixed-race ethnicity in the US). Standardized reliability (internal consistency) as indicated by Cronbach's alpha for MBI, GAD-7, PHQ-9 and SF-8 instruments were 0.73, 0.91, 0.85, and 0.84, respectively and were considered acceptable.

Based on whole model analysis with MANOVA, group (UCD or UP) was a significant predictor ($P <0.0001$; Wilks's Lambda $<0.0001$) influencing the scores on the survey instruments. Gender ($P = 0.264$), age ($P = 0.189$), and marital status ($P = 0.969$) were not significant predictors of scores on the survey instruments.

### Anxiety, burnout, and depression

Scores for GAD-7 were not different ($P = 0.126$) between UCD (score = 8; 95% CI, 7, 10) and UP (score = 9; 95% CI, 7, 12). The GAD-7 scores for both schools indicated moderate levels of anxiety. Score comparisons for the MBI between the two schools are summarized in Table 1. Students from UCD reported higher emotional exhaustion scores (33 vs 25; $P <0.0023$) compared to UP students. Scores for depersonalization ($P = 0.931$) and personal accomplishment ($P = 0.123$) were not different between the two schools. Scores for personal accomplishment for both schools indicated high levels of burnout whereas scores for depersonalization and emotional exhaustion indicated average to high levels of burnout. Scores (95% CI) for PHQ-9 were 8 (6, 9) and 10 (7, 12) for UCD and UP students, respectively. The PHQ-9 scores were not different ($P = 0.233$) between the two schools and indicated mild to moderate levels of depression. Strong correlation ($r = 0.74$) was present between GAD-7 and PHQ-9 scores.

### Quality of life

Quality of life dimension scores are summarized in Table 2. Scores for quality of life dimensions were below 50, except for the bodily pain dimension for UCD students. Similarly, scores

**Table 1. Comparison of scores (median and 95% confidence interval) of burnout with Maslach Burnout Inventory (MBI) between University of California (UCD) and University of Pretoria (UP) veterinary medical students.**

|  | UCD (N = 102) | UP (N = 74) | *P*-value |
|---|---|---|---|
| Emotional exhaustion (EE) | 33 (29, 35) | 25 (20, 28) | <0.0023 |
| Depersonalization (DP) | 7 (5, 9) | 7 (6, 9) | 0.931 |
| Personal accomplishment (PA) | 30.0 (28, 32) | 29.5 (27, 31) | 0.123 |

EE scores of ≤ 18, 19–26 and ≥27 indicate low, average, and high levels of burnout, respectively.

DP scores of ≤ 5, 6–9 and ≥10 indicate low, average, and high levels of burnout, respectively.

PA scores of ≥40, 39–34 and ≤33 indicate low, average, and high levels of burnout, respectively.

Row score comparisons between the 2 schools with *P* <0.05 are different.

for quality of life dimensions for UP students were below 50, except for the physical functioning dimension. The score for physical functioning dimension was higher (*P* = 0.039) in UP compared to UCD students. Students from UCD (50.3 ± 7.3) had higher scores (*P* = 0.02) for the bodily pain dimension compared to UP (48.5 ± 8.3) students. Mental component summary scores were below 50 for both UCD (36.5 ± 11.5) and UP (34.5 ± 12.3) but scores were not different (*P* = 0.418). Physical component summary score for UCD students was 49.8 ± 7.6 and was not different (*P* = 0.727) from UP students (50.8 ± 8.4). Strong correlations were present between mental component score summary and PHQ-9 scores (*r* = -0.765), and between mental component score summary and GAD-7 scores (*r* = -0.728).

## Discussion

Contrary to our hypothesis, study findings suggest that students from UCD and UP had comparable levels of anxiety, burnout, depression, and quality of life. Our findings are consistent with studies in US veterinary schools [5, 24] but there are no comparable studies available for South African veterinary students. In the US schools of veterinary medicine, risk factors associated with anxiety and depression are perceived poor physical health, unclear expectations in the curriculum, difficulty fitting in with peers, heavy academic workload, and homesickness [3, 4, 25, 26]. In contrast, the most frequent perceived contributors to student stress identified

**Table 2. Comparison of quality of life dimensions scores (mean ± standard deviation) with Short Form-8 (SF-8) scores between University of California (UCD) and University of Pretoria (UP) veterinary medical students.**

|  | UCD (N = 102) | UP (N = 74) | *P*-value |
|---|---|---|---|
| Role emotional (RE) | 38.7 ± 10.0 | 35.6 ± 10.6 | 0.114 |
| Mental health (MH) | 40.0 ± 7.7 | 42.0 ± 8.2 | 0.09 |
| Social functioning (SF) | 42.6 ± 9.4 | 39.6 ± 9.2 | 0.179 |
| Vitality (VT) | 45.0 ± 8.3 | 45.6 ± 7.6 | 0.795 |
| General health perception (GH) | 43.2 ± 9.9 | 44.4 ± 8.5 | 0.554 |
| Role physical (RP) | 47.8 ± 7.0 | 48.8 ± 7.3 | 0.443 |
| Physical functioning (PF) | 47.7 ± 7.1 | 50.2 ± 7.5 | 0.039 |
| Bodily pain (BP) | 50.3 ± 7.3 | 48.5 ± 8.3 | 0.020 |
| Mental component summary (MCS) | 36.5 ± 11.5 | 34.5 ± 12.3 | 0.418 |
| Physical component summary (PCS) | 49.8 ± 7.6 | 50.8 ± 8.4 | 0.727 |

Scores below 50 points (mean score = 50, standard deviation = 10) corresponded to deviations from normality and indicate a poorer quality of life, whereas scores above 50 points represented a better quality of life than that of the average adult American population [22]. Row score comparisons between the 2 schools with *P* <0.05 are different.

at UP during a recent study included academic pressure and heavy academic workload, electrical power failures, and isolation due to the remote location of the veterinary campus (unpublished data). Furthermore, UP has the only faculty of veterinary medicine in South Africa, a country with a population of > 55 million people (United Nations, World Population Prospects 2019, Office of the Director, DESA/Population division, New York, NY), thereby adding more scrutiny on its role in public service. Consequently, UP must justify the current and future resources to maintain and improve mental health and wellness for veterinary students.

The similarities in mental health and wellness concerns between the schools might be due to common stressors in any veterinary curriculum, including heavy academic workload. The results of our study indicate that recommendations and resources to improve mental health and wellness in the US might be applicable to South African veterinary medicine students. Specific strategies for improving mental health and wellness include curricular changes [3, 4], providing on-site resources for students' mental health [4], increasing awareness of mental health among students and faculty [4], and inclusion of mindfulness-based stress reduction intervention programs in the curriculum [27–31]. Although no comparable studies using the GAD-7 in veterinary students are available, the moderate levels of anxiety reported in our study is consistent with previous studies [3]. It should be noted that our study did not evaluate other potential risk factors for anxiety and depression including cultural [6, 7], social-economic [10], and racial factors [6, 8, 9], which have been reported in non-veterinary students and adult South Africans. Thus, further studies examining these risk factors for anxiety and depression are warranted in the South African veterinary student population.

Our study yielded levels of burnout consistent with previous studies at UCD [32]. Female students at UP (only one male student responded from UCD but was excluded) were likely to have higher scores of emotional exhaustion, indicating higher levels of stress consistent with previous studies in veterinary medical students at UCD [5]. It is important to note that the scores for personal accomplishment indicated high levels of burnout for students enrolled at both schools. The personal accomplishment subscale assesses feelings of competence and successful achievement in a student's schoolwork. Personal accomplishment and depersonalization determine emotional exhaustion [33], in contrast to studies by Maslach [34], which focused on emotional exhaustion. Thus, focusing on early signs of emotional exhaustion is not recommended because when the signs of emotional exhaustion appear, the burnout process is already underway [33]. To increase the sense of personal accomplishment and counteract burnout, specific training programs were recommended for employees in various fields [35]. These training programs include role-playing to provide success experiences (enactive mastery), models of performances (vicarious experiences), coaching and encouragement (verbal persuasion) [35]. Of these recommendations, coaching and encouragement are two approaches that can be incorporated into stress management classes in the veterinary curriculum.

The mild to moderate levels of depression reported in our study is consistent with previous studies in veterinary students. Gender was not a significant predictor of levels of depression, which is in contrast to previous studies that indicated that that female students experienced higher levels of depression compared to male students [4, 5]. A likely reason for this difference is the fewer number of male students (1 at UCD who was excluded, and 21 at UP) in our study. Thus, the results of this study only reflected the analysis of gender for the UP cohort. Although not significant predictors of mental health in our study, we had hypothesized that older or married students would report higher scores of depression due to additional stressors from responsibilities in their personal life. However, older students or married students might be more resilient in adapting to stressors from schoolwork as they rely on personal experiences.

Assessment of quality of life allows identification of the most compromised dimensions of well-being and therefore, the ability to intervene by providing resources for students. Quality of life assessments allow comparison of the students in our study to a reference US adult population, and therefore are interpreted as deviations from normality. Our study yielded lower quality of mental health in veterinary students which may be related to the stressors associated with being enrolled in veterinary school. In contrast, students from both schools exhibited better physical health likely due to the young age of our study population (<30 years). The physical component summary scores decrease with age [36]. The nature of the veterinary curriculum which includes physical and outdoor activities and working with animals may also have contributed to the better physical health. General factors found to decrease quality of life in medical students which are applicable to veterinary students include peer competition, unprepared teachers, excessive learning activities, harsh social realities, frustrations with the medical program, and insecurity regarding professional future [37]. In contrast, good teachers, classes with good didactic approaches, active learning methodologies, efficient time management, and meaningful relationships with family members, friends, and teachers are associated with improved quality of life [37]. While these factors can be addressed with curricular changes, quality of life in veterinary medical students may also be improved by inclusion of mindfulness-based stress reduction intervention programs [38].

## Implications of this study for collaborative future studies

The broader implications of this study include the importance of collecting data on mental health and wellness in veterinary programs during curricular implementation. Student mental health and wellness assessment will increase awareness among students and instructors of the impact of curricular revision, implementation, and evaluation. Furthermore, veterinary colleges should consider sharing resources and information on improving mental health and wellness during curricular revision and implementation.

Although our study is a preliminary study, the results serve to facilitate collaborative research between the two schools. Specific future large, longitudinal, research studies will focus on assessing social, racial, cultural, and economic factors associated with anxiety and depression. Furthermore, studies on impact of implementation and monitoring of mindfulness-based stress reduction intervention programs for veterinary students across all stages of the curriculum are warranted.

## Limitations of the study

The non-probability convenient sample size was relatively small consisting of a single class from each school; therefore, the results of this study cannot be generalized to populations other than those included in the sample because the sample is not representative of all veterinary students in the United States or Africa. The non-probability voluntary sample may produce a voluntary response bias as students who were more interested in the topic self-selected to participate in the study. Consequently, a nonresponse bias may also be produced where students choosing not to participate may have differed from the sample in several ways. This may result in an over or under representation of students from particular groups. External validity of the results in our study may be limited as data were self-reported and are prone to response bias, honesty and image management, introspective ability, understanding questions, rating scales, ordinal measures, and control of sample [39, 40]. Response bias is when a student responds in a particular way to an item regardless of what is assessed. Honesty and image management refers when a student may not respond honestly due to the nature of the question in order to manage how he/she will be perceived. Introspective ability addresses a student's lack

of ability to reflect accurately on one's self. Students may also vary in their ability to understand and interpret questions especially when dealing with abstract concepts, such as perceived level of stress. The value of each interval of a rating scale may be interpreted differently among students, as well as the distance between ordinal measures in a scale. Finally, students complete the survey at their own convenience, and the environment cannot be controlled. While our study did allow students to respond only once by soliciting surveys via individual emails, there is no control over the attention given to the survey by the student or if the indicated student completed the survey, thus we relied on the honesty of individual students.

Design, modification or change of the curriculum has an impact on student mental health and wellness. The effect of the curricula changes on mental health and wellness were not evaluated at both schools in our study to allow sufficient time for adjustments to the revised curricula. Holistic curricula evaluation, including assessment of mental health and wellness are planned for 2021 and 2022 academic years for UCD and UP, respectively.

Although the UP cohort had 13 students >26 years, age distribution in the 2 cohorts consisted of a higher proportion of students aged 20–25. Thus, the results from our study might be limited to this age group. Male participants were fewer for both schools with only 1 male respondent from UCD. Thus, analysis of gender comparisons was only applicable to the UP cohort. Further studies are required to evaluate the association between gender or age and scores of instruments assessing mental health and wellness in veterinary students from the two continents.

## Conclusion

Instruments for assessing anxiety, burnout, depression, and quality of life had acceptable reliability. Students from both schools in our study had moderate levels of anxiety, high levels of burnout, mild to moderate levels of depression, poor mental health, and good physical health. The results suggest that mental health and wellness concerns in South African veterinary students is comparable with concerns in the US. Recommendations and resources to improve mental health and wellness in the US might be applicable to a South African school of veterinary medicine.

## Acknowledgments

The authors thank the University of California, Davis School of Veterinary Medicine and University of Pretoria, Faculty of Veterinary Science students who took their time to complete the instruments.

## Author Contributions

**Conceptualization:** Munashe Chigerwe, Dietmar E. Holm, Karen A. Boudreaux.

**Data curation:** Munashe Chigerwe, El-Marie Mostert, Kate May, Karen A. Boudreaux.

**Formal analysis:** Munashe Chigerwe, Karen A. Boudreaux.

**Funding acquisition:** Dietmar E. Holm.

**Investigation:** Munashe Chigerwe, El-Marie Mostert, Kate May, Karen A. Boudreaux.

**Methodology:** Munashe Chigerwe, Karen A. Boudreaux.

**Project administration:** Karen A. Boudreaux.

**Writing – original draft:** Munashe Chigerwe.

**Writing – review & editing:** Munashe Chigerwe, Dietmar E. Holm, El-Marie Mostert, Karen A. Boudreaux.

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
