## [Decision Letter · Decision Letter 0]

26 Jun 2020

PONE-D-20-09786

Exploring issues surrounding mental health and wellbeing across two continents: A preliminary cross-sectional collaborative study between the University of California, Davis, and University of Pretoria

PLOS ONE

Dear Dr. Chigerwe,

Thank you for submitting your manuscript to PLOS ONE. After careful consideration, we feel that it has merit but does not fully meet PLOS ONE’s publication criteria as it currently stands. Therefore, we invite you to submit a revised version of the manuscript that addresses the points raised during the review process.

I received extensive feedback and I have carefully reviewed the manuscript as well. Overall, we thought that this manuscript presented an important, understudied topic that may generalize more broadly. Our reviewer offered some points of clarification and suggestions for edits that I would ask you to please address in a revision. I also suggest clarifying the statistical analyses and the choice to run univariate followed by multivariate tests. It seems like solely using a multivariate approach like MANOVA would be more appropriate for reducing the number of comparisons and capturing the likely correlations between subscales.

We look forward to receiving your revised manuscript.

Kind regards,

Rachel A. Annunziato, Ph.D.

Academic Editor

PLOS ONE

Journal Requirements:

2. During our internal checks, the in-house editorial staff noted that you conducted research or obtained samples in another country. Please check the relevant national regulations and laws applying to foreign researchers and state whether you obtained the required permits and approvals. Please address this in your ethics statement in both the manuscript and submission information.

"The study was funded in part by the University Capacity Development Grant of the Department of Higher Education and Training (DHET) of South Africa."

Reviewers' comments:

Reviewer's Responses to Questions

**Comments to the Author**

1. Is the manuscript technically sound, and do the data support the conclusions?

Reviewer #1: Yes

2. Has the statistical analysis been performed appropriately and rigorously? 

Reviewer #1: Unsure

3. Have the authors made all data underlying the findings in their manuscript fully available?

Reviewer #1: No

4. Is the manuscript presented in an intelligible fashion and written in standard English?

Reviewer #1: No

5. Review Comments to the Author

Reviewer #1: This preliminary research study examines and compares the mental health and wellness between veterinary students at the University of California Davis (UCD) and University of Pretoria (UP). Through measurements of anxiety, burnout, depression, and quality of life of students at comparable stages of their veterinary education, the authors found similar concerns for mental health and wellness between both schools. As such, they conclude that the recommended implementations to increase mental health and wellness in North American veterinary schools may be applicable to the University of Pretoria in South Africa. While these results were inconsistent with the authors’ original hypothesis, I expect that further research assessing the different cultural, social, racial and economic risk factors between North American and South African veterinary students will most likely have different implications for the recommendations that would be most effective for South African students.

The major strengths of the paper are contingent on the authors’ extensive knowledge of the topic and the pertinent existing research. The overall paper and conclusion are strengthened by the authors’ explication of their reasoning behind the choices of the data instruments and methods, as well as providing definitions and explanations for concepts that may not be familiar to individuals outside this specialty, such as those discussed in the “Limitations of the study” section (pages 17-18). This is important as it may help increase collaboration and attention to this topic from a cross-disciplinary perspective. Furthermore, it improves readability and reader comprehension. Another strength of the study is their extensive and honest section outlining the limitations of their research study and design and the implications of these for future studies.

However, with this being said, the extensive limitations of the study are what I perceive to be the major weakness of this paper. The various flaws of the study design, including the generalizability and external validity of results, most likely significantly impact the results and conclusions of this study. This may weaken its reproducibility and significance. Other major and minor issues are discussed below.

Major Issues

• The authors identify that mental health and wellness research is a topic of importance in North American veterinary schools (line 36). However, they fail to explain why it is important for this specific discipline. With a brief explanation of the importance of this and future research, the authors will provide readers and future researchers with a stronger impetus to perform further research studies, such as those recommended at the end (“Implications of this study for collaborative future studies” section on page 17).

• In the Introduction (lines 59-76), the authors identify certain curricular revisions that have been implemented in North American veterinary schools, with specific attention to those at UCD. To strengthen the association the authors make between mental health and wellness in veterinary students with modifications to and change of curriculum, a brief discussion of the success of current curriculum reforms would be helpful. Furthermore, the authors should consider briefly describing the similarities between the curriculum changes implemented at UCD and UP. Failure to describe the curriculum changes that have been implemented at UP is a significant issue that should be addressed to increase the strength of the authors’ conclusion.

Minor Issues

• While the authors exemplify extensive knowledge of the topic, there are many grammatical errors that make certain sections difficult to read and comprehend, specifically the common occurrence of run-on sentences. Specific attention should be given to the Introduction section.

• The authors identify the independent variables of the study, which are group, gender, marital status, and age. While they reference previous studies that found significant differences by gender in North America, further explanation for their choice to include marital status and age should be briefly discussed. Additionally, race would have been an easy descriptive statistic to obtain and incorporate into the research. This is an important factor to consider as there is significant evidence of racial disparities in mental health.

• It is important that the authors included a section on the current mental health and wellness support for veterinary students at UCD and UP. However, positioning this section as part of the Study Design and Methods section is confusing as it is merely background information and not relevant to how the study was conducted. As such, I advise that the authors move this section to be included in the Introduction for coherency and improved flow of the paper.

• Although the Discussion section is strong in discussing the results and their implications, it is not coherently structured. It would be helpful too to consider broader implications.

6. PLOS authors have the option to publish the peer review history of their article (what does this mean?). If published, this will include your full peer review and any attached files.

Reviewer #1: **Yes: **Mackenzie Connelly

---

## [Author Response · Author response to Decision Letter 0]

10 Aug 2020

31 July 2020

Dr. Annunziato

Academic Editor- PLOS ONE

Dear Dr., Annunziato,

We are pleased to submit the revised version of the manuscript entitled:

“A preliminary cross-sectional collaborative study between the University of California, Davis, and University of Pretoria” to PLOS ONE.

Thank you for giving us the opportunity to revise and resubmit this manuscript. We appreciate the time the reviewer spent on this manuscript and the detailed comments that they have provided. The manuscript has been revised to reflect your suggestions and the reviewer’s comments and suggestions. 

All authors have approved the final version of the revised manuscript.

We have responded specifically to each comment/suggestion below and highlighted any changes made to the manuscript in green (‘Manuscript with track changes’).

Academic Editor’s comments

I received extensive feedback and I have carefully reviewed the manuscript as well. Overall, we thought that this manuscript presented an important, understudied topic that may generalize more broadly. Our reviewer offered some points of clarification and suggestions for edits that I would ask you to please address in a revision. I also suggest clarifying the statistical analyses and the choice to run univariate followed by multivariate tests. It seems like solely using a multivariate approach like MANOVA would be more appropriate for reducing the number of comparisons and capturing the likely correlations between subscales.

AU Response: Thank you for the suggestion. We agree that analyzing the data using MANOVA is more appropriate compared to our previous approach that included adjustment of the P-values in the multiple comparisons. 

We re-analyzed the data using a one-way MANOVA. As a result of performing MANOVA, the following summarizes the changes (or no changes) in the results/conclusions:

1. Gender was not a significant predictor of any of the survey instrument scores in this revised version of the manuscript. In our first submission, gender was a significant predictor for emotional exhaustion (for UP students only because only 1 male student completed the survey at UCD) because we had analyzed the data as separate multiple regression models. Similar to our first submission, marital status and age were not significant predictors of scores on the survey instruments.

2. Group (UCD or UP) was a significant predictor of scores in at least one survey instrument, consistent with our first submission results/conclusions. Emotional exhaustion scores were higher for UCD students compared to UP students. This conclusion is similar to our first submission results. The P-value was different but still significant (P<0.0001, in our first submission versus P = 0.0023 in this revised version). Scores for depersonalization (P =0.931) and personal accomplishment (P = 0.123) were not different between the two schools, consistent with our first submission results.

3. Conclusions for differences between PHQ-9 and GAD-7 scores between UCD and UP remained unchanged, but the P-values changed, though still not significant. 

4. For SF-8: Social Functioning dimension score which was significantly different between the 2 schools was not significant (P = 0.179) with MANOVA. In contrast, Bodily Pain dimension (previously non-significant in the first submission) was significantly different (P=0.020) in this revised version.

The Physical Functioning (P = 0.039) was still significantly different between the two schools, consistent with the conclusion in the first submission. 

Although the p-values changed, Role Physical (P = 0.443), General Health (P=0.554), Vitality (P = 0.795), Role Emotion (P = 0.114), Mental Health (P = 0.09), Mental Component Summary (P=0.418), and Physical Component Summary (P =0.727) were not different between the schools, consistent with the conclusion in our first submission.

5. We revised the results, included analysis from MANOVA, and deleted results from multiple regression analysis for each survey instruments from our first submission. Please see results.

6. We also included the significant relevant correlations between the subscales. Please see results.

PLOS ONE style requirements

1. Please ensure that your manuscript meets PLOS ONE's style requirements, including those for file naming

AU Response: We followed all the style requirements and identified the revised editions of the manuscript as suggested by the journal.

2. During our internal checks, the in-house editorial staff noted that you conducted research or obtained samples in another country. Please check the relevant national regulations and laws applying to foreign researchers and state whether you obtained the required permits and approvals. Please address this in your ethics statement in both the manuscript and submission information.

AU Response: We provided a project approval number by the University of Pretoria. Please see Materials and Methods.

"The study was funded in part by the University Capacity Development Grant of the Department of Higher Education and Training (DHET) of South Africa."

AU Response: We deleted the funding source statement from the Acknowledgments and included it in our cover letter as suggested. We provided information on all the funding sources and included the statement; “There was no additional external funding received for this study” as suggested. We included the following funding source statement: “The study was funded by the University Capacity Development Grant of the Department of Higher Education and Training (DHET) of South Africa, and the UC Davis Faculty Discretionary Support funds. There was no additional external funding received for this study.”

AU response: The data presented in our study potentially contain sensitive identifying personal information such as, mental health data which have psychological risks to the respondents. We have provided contacts (non-authors of this manuscript) to which data requests maybe send. The contacts are the respective institutional representatives for the ethics committees:

University of California Davis:

Office of Research

IRB Administration

1850 Research Park Drive

Davis, CA 95618-6153

Phone: (530) 754-7679

Fax: (530) 754-7894

Email: ORExecutiveMgtAsst@ad3.ucdavis.edu

University of Pretoria: 

Deputy Dean, Postgraduate and Research Ethics

Faculty of Humanities

Email: PGHumanities@up.ac.za

AU response: Please see our response above, in 4(a).

Reviewer #1

General response: Thank you for taking time to review our manuscript. To make it easier for you to recognize the changes we made to the manuscript, we highlighted the changes in green.

Comments to the Author

1. Is the manuscript technically sound, and do the data support the conclusions?

Reviewer #1: Yes

AU response: Thank you.

2. Has the statistical analysis been performed appropriately and rigorously? 

Reviewer #1: Unsure

AU response: We considered recommendations from the Academic Editor, specifically to consider MANOVA (multivariate analysis of variance) rather than assessing the difference in scores for each survey instruments as single multivariate regression analysis. We clarified this in the materials and methods. We changed the results accordingly.

3. Have the authors made all data underlying the findings in their manuscript fully available?

Reviewer #1: No

AU response: The data presented in our study potentially contain sensitive identifying personal information such as mental health data which have psychological risks for the respondents. We have provided contacts to which data requests maybe sent. The contacts are the respective institutional representatives for the ethics committees:

University of California Davis:

Office of Research

IRB Administration

1850 Research Park Drive

Davis, CA 95618-6153

Phone: (530) 754-7679

Fax: (530) 754-7894

Email: ORExecutiveMgtAsst@ad3.ucdavis.edu

University of Pretoria: 

Deputy Dean, Postgraduate and Research Ethics

Faculty of Humanities

Email: PGHumanities@up.ac.za

4. Is the manuscript presented in an intelligible fashion and written in standard English?

Reviewer #1: No

AU response: We revised the manuscript and checked for any grammatical errors to the best of our ability.

5. Review Comments to the Author

Reviewer #1: This preliminary research study examines and compares the mental health and wellness between veterinary students at the University of California Davis (UCD) and University of Pretoria (UP). Through measurements of anxiety, burnout, depression, and quality of life of students at comparable stages of their veterinary education, the authors found similar concerns for mental health and wellness between both schools. As such, they conclude that the recommended implementations to increase mental health and wellness in North American veterinary schools may be applicable to the University of Pretoria in South Africa. While these results were inconsistent with the authors’ original hypothesis, I expect that further research assessing the different cultural, social, racial and economic risk factors between North American and South African veterinary students will most likely have different implications for the recommendations that would be most effective for South African students.

The major strengths of the paper are contingent on the authors’ extensive knowledge of the topic and the pertinent existing research. The overall paper and conclusion are strengthened by the authors’ explication of their reasoning behind the choices of the data instruments and methods, as well as providing definitions and explanations for concepts that may not be familiar to individuals outside this specialty, such as those discussed in the “Limitations of the study” section (pages 17-18). This is important as it may help increase collaboration and attention to this topic from a cross-disciplinary perspective. Furthermore, it improves readability and reader comprehension. Another strength of the study is their extensive and honest section outlining the limitations of their research study and design and the implications of these for future studies.

However, with this being said, the extensive limitations of the study are what I perceive to be the major weakness of this paper. The various flaws of the study design, including the generalizability and external validity of results, most likely significantly impact the results and conclusions of this study. This may weaken its reproducibility and significance. Other major and minor issues are discussed below.

AU response: Thank you for your succinct summary of the strength and weaknesses of our study. We agree with your comment that further longitudinal studies results might differ from this current study particularly when cultural, social, racial, and economic factors are considered. While all these factors have a potential influence on the mental health, the degree to which they affect mental health among veterinary students might be different between the schools. We plan to pursue longitudinal studies with intervention strategies at both schools.

Major Issues

1. The authors identify that mental health and wellness research is a topic of importance in North American veterinary schools (line 36). However, they fail to explain why it is important for this specific discipline. With a brief explanation of the importance of this and future research, the authors will provide readers and future researchers with a stronger impetus to perform further research studies, such as those recommended at the end (“Implications of this study for collaborative future studies” section on page 17).

AU response: We added relevant statistics from recent studies to provide more context on mental health and wellness in North American veterinary students compared to peers in other programs, such as medical students. We added the information in the introduction and cited relevant references. Please see the introduction.

2. In the Introduction (lines 59-76), the authors identify certain curricular revisions that have been implemented in North American veterinary schools, with specific attention to those at UCD. To strengthen the association the authors make between mental health and wellness in veterinary students with modifications to and change of curriculum, a brief discussion of the success of current curriculum reforms would be helpful. Furthermore, the authors should consider briefly describing the similarities between the curriculum changes implemented at UCD and UP. Failure to describe the curriculum changes that have been implemented at UP is a significant issue that should be addressed to increase the strength of the authors’ conclusion.

AU response: The revised curricula were implemented in 2011 and 2016, at UCD and UP, respectively. As you pointed out, the revised curricula will have an impact on student mental health and wellness. Curricular evaluation at UCD is planned in 2021 (5 years after the first class on the new curriculum graduated) and 2022 at University of Pretoria. In response to your comment, we added the lack of data regarding the impact of the curricular change on mental and wellness as limitation of our study. Please see changes under ‘Limitations of the study’.

While not numerous, we included the similarities between the drivers for curricular changes between UCD and UP. Please see introduction.

Minor Issues

1.While the authors exemplify extensive knowledge of the topic, there are many grammatical errors that make certain sections difficult to read and comprehend, specifically the common occurrence of run-on sentences. Specific attention should be given to the Introduction section.

AU response: We revised the introduction and split the run-on sentences where necessary but did our best to maintain flow of the manuscript. 

2. The authors identify the independent variables of the study, which are group, gender, marital status, and age. While they reference previous studies that found significant differences by gender in North America, further explanation for their choice to include marital status and age should be briefly discussed. Additionally, race would have been an easy descriptive statistic to obtain and incorporate into the research. This is an important factor to consider as there is significant evidence of racial disparities in mental health.

AU response: As indicated in our first submission, most students from UCD are 25 years and younger (at this stage of the curriculum) and this is expected in North American veterinary schools. Only 1 student was aged 26-30 from UCD 13 students from UP were aged 26-30 hence no comparison was made between the schools. In response to your comment, we did the following:

1. Discussed potential reasons why we included marital status and age variables for predicting scores. Please see Discussion.

2. We provided descriptive statistics of the racial composition of the 2 classes included in this study. Please see Results. 

3. It is important that the authors included a section on the current mental health and wellness support for veterinary students at UCD and UP. However, positioning this section as part of the Study Design and Methods section is confusing as it is merely background information and not relevant to how the study was conducted. As such, I advise that the authors move this section to be included in the Introduction for coherency and improved flow of the paper.

AU response: Thank you for the suggestion. We moved the material on mental and health support for UCD and UP to the introduction. We provided sub-titles in the introduction to improve the flow of the manuscript. Please see changes in the introduction.

4. Although the Discussion section is strong in discussing the results and their implications, it is not coherently structured. It would be helpful too to consider broader implications.

AU response: We expanded the discussion to include broader implications of the study. Specifically, we discussed the impact of the study results relating to the following:

1. The importance of collecting data and monitoring mental health and wellbeing to increase awareness by students and faculty.

2. Sharing and provision of resources between schools to monitor mental health and wellness. 

To improve the flow of the Discussion, we included this information under “Implications of the study for future studies”. We think the information fits better in this part of the discussion as it summarizes the practical implications of the study.

Please see Discussion for the additions.

---

## [Decision Letter · Decision Letter 1]

18 Sep 2020

PONE-D-20-09786R1

Exploring issues surrounding mental health and wellbeing across two continents: A preliminary cross-sectional collaborative study between the University of California, Davis, and University of Pretoria

PLOS ONE

Dear Dr. Chigerwe,

Thank you for submitting your manuscript to PLOS ONE. After careful consideration, we feel that it has merit but does not fully meet PLOS ONE’s publication criteria as it currently stands. Therefore, we invite you to submit a revised version of the manuscript that addresses the points raised during the review process.

We look forward to receiving your revised manuscript.

Kind regards,

Rachel A. Annunziato, Ph.D.

Academic Editor

PLOS ONE

Reviewers' comments:

Reviewer's Responses to Questions

**Comments to the Author**

1. If the authors have adequately addressed your comments raised in a previous round of review and you feel that this manuscript is now acceptable for publication, you may indicate that here to bypass the “Comments to the Author” section, enter your conflict of interest statement in the “Confidential to Editor” section, and submit your "Accept" recommendation.

Reviewer #1: (No Response)

2. Is the manuscript technically sound, and do the data support the conclusions?

Reviewer #1: Yes

3. Has the statistical analysis been performed appropriately and rigorously? 

Reviewer #1: Yes

4. Have the authors made all data underlying the findings in their manuscript fully available?

Reviewer #1: Yes

5. Is the manuscript presented in an intelligible fashion and written in standard English?

Reviewer #1: No

6. Review Comments to the Author

Reviewer #1: The authors made many revisions that have significantly strengthened their paper. Aside from the strengths discussed in their first submission, one strong addition was the inclusion of past studies that explicate the importance of studying mental health and wellness in veterinary school students. This will provide a further impetus for future studies to be conducted. The manuscript is further strengthened by the addition of the “Data Analysis” subsection, format changes to improve the paper’s readability, as well as the explanation of the specific independent variables that were studied.

Major Revisions

• The authors state that the factors associated with good mental health and well-being are different between the two countries. They hypothesize that the levels of mental health and wellness would be different between the two schools. However, in the introduction, the authors also hypothesize that the approaches to improve mental health and well-being in North America may be applicable South African veterinary schools. The authors should include a brief explanation as to why they hypothesized this despite the many differences.

• The authors state that they are assessing differences between North American and South African veterinary school students. However, all of the data provided, as well as the study conducted, are only applicable to the United States. Therefore, I recommend revising this inaccuracy by replacing “North America(n)” with “the United States of America” or “America(n),” unless the authors can provide statistics that indicate there are comparable mental health and wellness concerns in both Canada and Mexico.

• Throughout the paper, the authors switch between “South Africa(n)” and “Africa(n).” 

Minor Revisions

• I recommend moving the primary research question stated in lines 141-143 to the abstract.

• The “Current mental health and wellness support for veterinary students” section is important. While they discuss the UCD “Wellness Center” and the programs it offers, the pathway to UP Counselling Services is less clear. I suggest briefly stating the programs it offers, if the information is available, to demonstrate consistency.

• To improve flow, readability and comprehension of the “Results” section, I suggest putting each school’s information consecutively. For example, state ages of respondents for UP followed by the ages of respondents for UCD. Or create separate paragraphs for each school.

• Although very minor, I suggest creating a new paragraph on line 93 to separate the background information discussion of North American and South African veterinary schools.

• The authors made many corrections to grammatical errors present in their first submission. However, there are still many run-on sentences and misuses of commas. I recommend revising these grammatical errors to improve readability and comprehension.

7. PLOS authors have the option to publish the peer review history of their article (what does this mean?). If published, this will include your full peer review and any attached files.

Reviewer #1: **No**

---

## [Author Response · Author response to Decision Letter 1]

5 Oct 2020

Reviewer #1: The authors made many revisions that have significantly strengthened their paper. Aside from the strengths discussed in their first submission, one strong addition was the inclusion of past studies that explicate the importance of studying mental health and wellness in veterinary school students. This will provide a further impetus for future studies to be conducted. The manuscript is further strengthened by the addition of the “Data Analysis” subsection, format changes to improve the paper’s readability, as well as the explanation of the specific independent variables that were studied.

Response

Thank you for taking time to review our manuscript. We responded point-by-point to your suggestions and comments. To make it easier for you to recognize the changes we made in the manuscript, we highlighted the changes in green where necessary.

Major Revisions

• The authors state that the factors associated with good mental health and well-being are different between the two countries. They hypothesize that the levels of mental health and wellness would be different between the two schools. However, in the introduction, the authors also hypothesize that the approaches to improve mental health and well-being in North America may be applicable South African veterinary schools. The authors should include a brief explanation as to why they hypothesized this despite the many differences.

Response

We hypothesized that the levels of mental health and wellness would be different based on the different stressors for being in veterinary school between the two different continents. Different stressors that will make approaches to mental health and wellness different between the schools maybe be stressors like powercuts (reducing study time), and isolation of the veterinary school in South Africa (there is only one veterinary school in South Africa); whereas perceived poor physical health and unclear expectations of the curriculum maybe be major stressors at North American schools. However, approaches to improving mental health maybe similar for stressors that are common to being in veterinary school such as heavy academic workload, and lack of racial diversity in class composition.

In response to your comment we clarified the statement in the introduction to indicate that although there are different stressors between the schools, improvement of mental health and wellness maybe applicable towards stressors that are common to both veterinary school.

• The authors state that they are assessing differences between North American and South African veterinary school students. However, all of the data provided, as well as the study conducted, are only applicable to the United States. Therefore, I recommend revising this inaccuracy by replacing “North America(n)” with “the United States of America” or “America(n),” unless the authors can provide statistics that indicate there are comparable mental health and wellness concerns in both Canada and Mexico.

Response

Thank you. We changed the phrase from “North America” to the “United States of America (US)” throughout the manuscript.

• Throughout the paper, the authors switch between “South Africa(n)” and “Africa(n).” 

Response

We changed the phrase to “South African” throughout the manuscript because our study focused on South African students.

Minor Revisions

• I recommend moving the primary research question stated in lines 141-143 to the abstract.

Response

We added the primary research question to the abstract. To maintain flow of the manuscript, we left the primary research question in the introduction under the sub-title “Current Study” as well.

• The “Current mental health and wellness support for veterinary students” section is important. While they discuss the UCD “Wellness Center” and the programs it offers, the pathway to UP Counselling Services is less clear. I suggest briefly stating the programs it offers, if the information is available, to demonstrate consistency.

Response

The information indicated in the manuscript for UP Counselling Services is how the program is set-up currently and there is no additional information regarding its organization. Most of the staff administering the program are part-time except the full-time faculty advisor. However, the full-time faculty advisor provides mentoring and life coaching, but is not qualified/licensed to provide psychotherapy. The program at UP is still growing at the moment.

• To improve flow, readability and comprehension of the “Results” section, I suggest putting each school’s information consecutively. For example, state ages of respondents for UP followed by the ages of respondents for UCD. Or create separate paragraphs for each school.

Response

Thank you. We re-organized the results as suggested. Please see the results.

• Although very minor, I suggest creating a new paragraph on line 93 to separate the background information discussion of North American and South African veterinary schools.

Response

We created a new paragraph to separate background information on US veterinary schools and the South African school. Please see introduction.

• The authors made many corrections to grammatical errors present in their first submission. However, there are still many run-on sentences and misuses of commas. I recommend revising these grammatical errors to improve readability and comprehension.

Response

Thank you for the comment. We revised the manuscript and divided the run-on sentences when necessary. We also removed inappropriate commas when necessary using Microsoft Word grammar and spell-check.

---

## [Editor Report · Decision Letter 2]

13 Oct 2020

Exploring issues surrounding mental health and wellbeing across two continents: A preliminary cross-sectional collaborative study between the University of California, Davis, and University of Pretoria

PONE-D-20-09786R2

Dear Dr. Chigerwe,

We’re pleased to inform you that your manuscript has been judged scientifically suitable for publication and will be formally accepted for publication once it meets all outstanding technical requirements.

Kind regards,

Rachel A. Annunziato, Ph.D.

Academic Editor

PLOS ONE
---

## [Editor Report · Acceptance letter]

15 Oct 2020

PONE-D-20-09786R2 

Exploring issues surrounding mental health and wellbeing across two continents: A preliminary cross-sectional collaborative study between the University of California, Davis, and University of Pretoria 

Dear Dr. Chigerwe:

I'm pleased to inform you that your manuscript has been deemed suitable for publication in PLOS ONE. Congratulations! Your manuscript is now with our production department. 

Kind regards, 

on behalf of

Dr. Rachel A. Annunziato 

Academic Editor

PLOS ONE